# The Rhizobial Type 3 Secretion System: The Dr. Jekyll and Mr. Hyde in the Rhizobium–Legume Symbiosis

**DOI:** 10.3390/ijms231911089

**Published:** 2022-09-21

**Authors:** Irene Jiménez-Guerrero, Carlos Medina, José María Vinardell, Francisco Javier Ollero, Francisco Javier López-Baena

**Affiliations:** Departamento de Microbiología, Universidad de Sevilla, Avenida de Reina Mercedes, 6, 41012 Sevilla, Spain

**Keywords:** type 3 secretion system, T3SS, effector, symbiosis, rhizobium

## Abstract

Rhizobia are soil bacteria that can establish a symbiotic association with legumes. As a result, plant nodules are formed on the roots of the host plants where rhizobia differentiate to bacteroids capable of fixing atmospheric nitrogen into ammonia. This ammonia is transferred to the plant in exchange of a carbon source and an appropriate environment for bacterial survival. This process is subjected to a tight regulation with several checkpoints to allow the progression of the infection or its restriction. The type 3 secretion system (T3SS) is a secretory system that injects proteins, called effectors (T3E), directly into the cytoplasm of the host cell, altering host pathways or suppressing host defense responses. This secretion system is not present in all rhizobia but its role in symbiosis is crucial for some symbiotic associations, showing two possible faces as Dr. Jekyll and Mr. Hyde: it can be completely necessary for the formation of nodules, or it can block nodulation in different legume species/cultivars. In this review, we compile all the information currently available about the effects of different rhizobial effectors on plant symbiotic phenotypes. These phenotypes are diverse and highlight the importance of the T3SS in certain rhizobium–legume symbioses.

## 1. Introduction

A group of soil α- and β-proteobacteria, called rhizobia, and legumes can establish a nitrogen-fixing symbiotic interaction [1,2]. Rhizobia live as saprophytes in soils but, in the presence of an appropriate legume partner, can transition to an endosymbiotic form, called a bacteroid, inside nodules, which are new organs formed by the plant in response to the presence of these bacteria [3]. Bacteroids, in contrast to free-living rhizobia, express genes required for nitrogen fixation. Thus, bacteroids provide an assimilable form of nitrogen (ammonia) to the plant, which, in turn, feeds (supplying a carbon and energy source) and houses them inside a stable environment.

The symbiotic interaction between rhizobia and legumes involves bacterial colonization of the rhizosphere, infection of legume roots, and invasion of the symbiotic nodule plant cells, which eventually host thousands of bacteroids. This interaction is specific: each rhizobial strain can nodulate (i.e., establish a nitrogen-fixing symbiosis) with a definite set of legumes (known as the host-range), which can vary from a few legumes to more than one hundred of the legume genera [3,4,5]. This specificity relies in two main (and related) facts. On the one hand, rhizobia must overcome several checkpoints during root colonization and infection and nodule invasion, synthesizing the right molecular signals that will be recognized by the plant to assure the progression of symbiosis [1,2,5,6,7,8]. The nature and mode of action of these rhizobial molecular signals have been mainly studied in a limited number of model plants and model rhizobia, and many variations can be found when other symbiotic couples are analyzed [9]. On the other hand, rhizobia, since they are infecting and invading plant tissues, must suppress and overcome plant immune responses [10,11].

As we will briefly describe below, different rhizobial molecular signals participate in symbiosis: nod factors, surface polysaccharides and effector proteins delivered by specialized secretion systems (Figure 1). However, this review focuses on the effector proteins delivered by rhizobial type 3 secretion systems (T3E) and provides a description of the different T3E proteins described so far as well as an exhaustive compilation of their role (positive, neutral, or negative) in symbiosis with specific counterparts.

### 1.1. Rhizobial Nod Factors and Surface Polysaccharides Are Key Signal Molecules in Most Symbiotic Interactions with Legumes

Flavonoids present in legume root exudates and, when appropriate, interact with rhizobial LysR regulatory proteins (NodD proteins), promoting the expression of rhizobial nodulation genes (*nod* genes), which results in the biosynthesis and secretion of molecular signals called nodulation factors (NF) or lipochitooligosaccharides (LCO) [4,8,12]. NF, which are N-acetyl-glucosamine oligosaccharides harboring different molecular decorations, when compatible, are recognized by plant LysM receptors, allowing root infection and colonization of the symbiotic cells within nodules [1,2]. Perception of NF by plant LysM receptors triggers a very complex regulatory cascade that has been detailed in several recent reviews [1,6,9,13]. Therefore, the two above-mentioned molecular interaction/recognition events (flavonoids–NodD and NF–LysM receptors) are key factors for the specificity of the rhizobium–legume interaction.

Rhizobia must penetrate the root epidermal barrier to reach the developing nodule. To date, two different mechanisms have been described for this infection event: entry by infection threads (IT) or through intercellular infection [14,15]. These mechanisms take place in approximately 75% and 25% of the legume genera, respectively [15]. Root hair IT-mediated infection, which is considered a mode of infection more evolved than intercellular infection, is an NF-dependent process: perception of these signal molecules by root hair receptors is required for root hair deformation and the entrapping of rhizobial cells, and for root hair plasmatic membrane invagination, which initiates the formation of the tubular structure that will elongate and branch, as the way for rhizobia to reach nodule primordia. However, NF are also required in later steps of the nodulation process [16,17,18].

In contrast to IT formation, intercellular infection remains poorly understood [14,15,19]. In some legumes, such as *Sesbania* spp. and *Lotus* spp., the natural mode of infection is through IT, but intercellular infection occurs under certain growth conditions, such as flooding, for *Sesbania* spp., or interacting with specific rhizobial partners, such as the symbiosis between *Lotus burttii* and *Sinorhizobium fredii* HH103 [20,21]. In other legumes, such as *Arachis hypogaea* and *Aeschynomene evenia*, intercellular infection is the natural method of rhizobial entry. In any case, intercellular infection can develop via a crack-entry mechanism, typically through epidermal fissures of emerging lateral roots, such as in *Sesbania*, *Aeschynomene* or *Arachis*, or in root zones where massive root hair curling and twisting takes place, such as in the interaction of strain IRBG74 with *L. japonicus* [19]. Several studies indicate NF are required in some but not all the intercellular infection processes described so far [15,19,22,23,24].

Rhizobial NF synthesis is a very complex process, and the regulatory circuits involved vary from one rhizobial species (or strain) to another [4,5,8]. In fact, there are several different possibilities of regulation: (i) the number of different NodD proteins may vary from one to five and different NodD proteins may recognize different signal molecules from the plant [4,16,25,26]. Moreover, *R. tropici* CIAT899 can induce NF synthesis not only in response to flavonoids but also under osmotic stresses [27,28,29]; (ii) in some cases, some NodD variants may even have a negative effect on NF production [4,30,31]; (iii) in *Sinorhizobium* spp. and *R. leguminosarum*, the global regulator NolR acts as a repressor of NF production [31,32,33,34]; and (iv) another LysR regulatory protein, SyrM, is involved in positive and negative regulation of NF production in *S. meliloti* and *S. fredii*, respectively [35,36,37].

As mentioned above, NF are important for specificity in rhizobium–legume symbioses, since in most cases infection will only proceed when appropriate NF are recognized by plant receptors. However, it is not only a question of the specific NF produced but also of its quantity. This is the case of *L. burttii*, which is infected by IT or by intercellular infection in its interaction with its natural endosymbiont *Mesorhizobium loti* and with *S. fredii* HH103, respectively [21]. In this legume, overproduction of NF by HH103, caused by the inactivation of the transcriptional regulators NodD2, NolR, or SyrM, likely accounts for the switch of the mode of infection to IT formation (observed for HH103 mutants in these regulators) and gaining effective nodulation with *L. japonicus* and *Phaseolus vulgaris* [31,37,38].

Several rhizobial surface polysaccharides (RSP) are also important in symbiosis [5,7,8]. In general, RSP can act either as signal molecules required for progression of the symbiosis and/or as protective agents against plant defense responses [7,8,39]. Four RSP have been deeply studied in relation to the symbiotic interaction with legumes: cyclin glucans (CG), lipopolysaccharides (LPS), and exopolysaccharides (EPS). The latter, which are homo- or heteromeric acidic polysaccharides secreted to the extracellular environment and located on the cell surface but with little or no cell association, show a complex regulation that is frequently influenced by flavonoids, either positively or negatively depending on the rhizobial strain. For example, *nod* gene-inducing flavonoids and NodD stimulate or repress EPS production in *R. leguminosarum* bv. *trifolii* and *S. fredii* HH103, respectively [40,41]. In symbioses with legumes forming indeterminate nodules, such as *Trifolium* spp. and *Medicago* spp., EPS-derived oligosaccharides have a signaling role in root infection and nodule invasion [7,8,39]. In interactions with determinate nodule-forming legumes, EPS has typically been considered dispensable, and even slightly detrimental in the interaction between *S. fredii* HH103 and soybean [7,8,39,42]; however, recent works carried out in the symbiotic couple *M. loti*-*L. japonicus* have revealed a more complex situation [43,44,45,46]. In this interaction, recognition of *M. loti* R7A NF by plant receptors (NFR1 and NFR5) triggers a regulatory cascade leading to the early steps of infection and nodule development, including the expression of EPR3, which functions as an EPS receptor that works in different steps of the symbiotic interaction, so that EPS recognition is reiterated along the nodulation process. *M. loti* R7A EPS is therefore recognized as adequate by this receptor, allowing successful infection and colonization.

### 1.2. Some Rhizobia Can Deliver Effector Proteins into Their Hosts through Bacterial Secretion Systems to Counteract Plant Immune Responses

Pathogenic or mutualistic Gram-negative bacteria that interact with eukaryotic organisms can deliver effector proteins into the cytosol of their host cells through specialized secretion systems such as the Type 3 (T3SS), Type 4 (T4SS), and Type 6 (T6SS) secretion systems [47,48,49,50]. Although these effectors have very different biochemical activities, their main function is to suppress plant defense responses to facilitate bacterial infection and survival inside the host.

Some rhizobial strains, but not all, use secretion systems to deliver these effector proteins. The most extended secretion system among rhizobia is the T3SS, which is present in some strains of the *Mesorhizobium*, *Bradyrhizobium*, *Rhizobium*, and *Sinorhizobium* genera, its presence being predominant in some cases such as in *Bradyrhizobium* spp. and *S. fredii* and scarce in *S. meliloti* [8,49]. The presence of symbiotic T4SS and T6SS appears to be less extended in rhizobia, at least in the strains characterized so far [51]. A symbiotic T4SS has been described in *M. loti* R7A and many *S. meliloti* and *S. medicae* strains [52,53]. The symbiotic effect, either neutral, positive, or negative, of mutations in the T4SS clearly depends on the symbiotic host, as demonstrated with T4SS mutants of *S. meliloti* and *S. medicae* in symbiosis with different genotypes of *Medicago* spp. [53]. Similarly, *M. loti* T4SS mutants showed delayed nodulation on *L. corniculatus* but extended host-range to *Leucaena leucocephala* [52].

The T6SS is mostly used by Gram-negative bacteria as a killing nanomachine devoted to outcompeting surrounding bacteria [48]. However, this secretion system is present in some rhizobia in which it acts as a determinant of host compatibility. It has been found in different strains belonging to the *Bradyrhizobium*, *Mesorhizobium*, and *Rhizobium* genera, as well as in *S. fredii* USDA257. In *R. leguminosarum*, the T6SS seems to restrict host compatibility since mutants in this secretion system gain the ability to effectively nodulate new hosts, whereas in *R. etli* Mim1 and *Bradyrhizobium* sp. LmicA16 it exerts a positive role in nodulation with *Phaseolus* spp. and *Lupinus* spp., respectively [54,55,56].

In the case of bacterial pathogens, the role of protein secretion systems is generally clear, whereas for beneficial bacteria the functions of such kinds of system remain in some cases elusive [10,11,51,57]. Different plant receptors recognize conserved motifs present on the surface of microbes/pathogens, which are collectively called microbe/pathogen-associated molecular patterns (MAMP/PAMP). This recognition triggers a rapid cell response called MAMP/PAMP-triggered immunity (MTI/PTI). Typical surface components that can induce this defense response are flagella and surface polysaccharides. In the case of rhizobia, flagella lack MAMP activity [10,57]. Instead, surface polysaccharides, mainly EPS and LPS, appear to play important roles in suppressing plant immune responses, although it is not clear yet how they inhibit plant defenses and whether their action is general or restricted to a specific symbiotic couple [8,10,57]. It is noteworthy that one of the responses mediated by MTI, the production of reactive oxygen species (ROS), also acts as mechanism important for symbiosis progression, and RSP have been shown to protect against these ROS [9,10].

In *L. japonicus* and *M. truncatula*, MAMP chitin oligomers are perceived by different, but related, LysM receptors other than NF receptors. This fact allows discrimination of chitin oligomers from pathogens or symbionts [58]. Moreover, it has been found that NF can partially suppress MTI, both in legumes and nonlegumes such as the model plant *A. thaliana* [59], although other studies showed opposite results in which NF may promote plant defense reactions [57]. Interestingly, T3SS effectors (T3E) can also suppress this early defense response to promote infection, as in the case of the *S. fredii* HH103-soybean interaction [60]. In summary, an early MTI response, which is quickly suppressed, appears to be necessary for optimal rhizobium–legume symbiotic interaction [10,57].

On the other hand, rhizobial T3E can be very specifically perceived by plant intracellular receptors, with very different results: either allowing symbiosis progression, or triggering a very robust immune response called effector-triggered immunity (ETI), which blocks nodulation [10,51]. Because of this, the set of effectors of each specific rhizobial strain delivered by T3/T4/T6SS may be a key feature for the success or fail of the symbiotic interaction with each specific legume.

## 2. The Symbiotic Type 3 Secretion System

Among all bacterial protein secretion systems, the T3SS is probably the best characterized. The T3SS is a nanomachine present in many Gram-negative bacteria that delivers proteins into the cytosol of eukaryotic host cells to, in most cases, manipulate their functions [61]. Its structure is composed of a set of proteins that spans the inner and outer membranes that serves as a basal complex to polymerize the extracellular component of the system, a long hollow tube of pilins that culminates in a translocon that recognizes host cell membranes where it forms a pore for protein delivery. The whole complex resembles a syringe that injects effector proteins directly from the cytosol of the bacterium to the cytosol of host cell. Although this system has been mainly studied in animal pathogens such as *Salmonella*, *Yersinia*, or *Escherichia coli* [62], its presence in numerous phytopathogens, such as strains of *Pseudomonas*, *Xanthomonas* or *Ralstonia*, has also been explored in depth [63]. Due to the obvious differences among animal and plant host-cell surfaces, T3SS reveals structural differences in the bacteria invading each kind of host, being longer and thinner in the case of plant-interacting bacteria [49]. Analogously, T3SS shows genetic differences among plant-interacting bacteria whether they are beneficial or pathogenic based on gene composition, arrangement, and transcriptional regulation [64].

In rhizobia, genes coding for the T3SS are usually contained on symbiotic plasmids (pSym) or in symbiotic islands in the chromosome. Genes encoding the core machinery are named *rhc* (Rhizobium conserved) and the secreted proteins (effectors, pilins, and proteins forming the translocon) are known as nodulation outer proteins (Nops) [65]. The *rhc* loci are quite conserved in most rhizobia and are clustered in regions ranging from 22 to 50 Kb, while genes encoding Nops can also be found dispersed throughout the genome. These Nops sometimes show similarities with T3E secreted by (phyto) pathogens suggesting, in some cases, a similar role in the host cell and even a common ancestor. However, other T3E can be considered *Rhizobium*-specific. In these cases, it is likely that their function is more associated with specific stages of the symbiotic process [66].

As part of the phylogenetic analysis, four subgroups of rhizobial *rhc* regions have been identified, where only the Rhc-I (containing members of *S. fredii*, *M. loti*, and *B. japonicum*) has been directly associated with symbiosis [64]. Rhc-II group is restricted to different *Sinorhizobium* strains (such as NGR234, HH103, and USDA257) but its role is still unclear. The Rhc-III subgroup is characterized by a completely different genetic organization compared to other rhizobia and includes strains of *R. etli*, *R. leguminosarum*, and some strains of *R. rhizogenes* or *R. radiobacter*. Finally, the β-Rhc gene cluster is represented only by the β-proteobacterium *Cupriavidus taiwanensis* that harbors a non-canonical genetic organization [64]. Very interestingly, rhizobial T3SS expression is co-regulated with NF production; it is activated by NodD and flavonoids through the induction of the *ttsI* gene, whose encoded product acts as the positive transcriptional regulator of genes coding for the T3SS apparatus and T3E [67,68]. A recent work showed that T3SS is absolutely required for genistein-induced surface motility in *S. fredii* HH103, revealing a new and unexpected function of T3SS in rhizobia [69].

## 3. Rhizobial Type 3 Secretion System Effectors

Although traditionally the assignment of the term “effector” to a putative secreted protein was based on its regulation through NodD and TtsI together with inducer flavonoids, and its translocation to a host cell, the new advances in the combination of multi-omics approaches (transcriptome and quantitative shotgun proteome analysis) with the experimental validation of obtained candidates will provide an ample amount of rhizobial T3E as recently described [70]. However, an interesting observation is that while pathogens secrete a plethora of effectors to overcome the different defense responses of target plants (as in the case of *P. syringae*, which secretes more than 80 effectors), rhizobia have limited their arsenal to a much lower number, especially in *Sinorhizobium* strains, which has about ten T3E [49,71]. A question arises from this fact: are those rhizobial T3E and their functions very closely related and perfectly coupled to develop the symbiotic process?

Even though several rhizobial T3Es have been identified, to date, only a restricted number of those T3Es have been described and their roles validated in plants (Table 1).

Rhizobial T3E, as well as in pathogens, can be composed by different modules with specific functional domains, and targets a single or diverse subcellular localizations into the plant cell [47,103,112]. Very interestingly, a great number of the studied T3E localize to the plant nucleus, where they can develop their functions. These are the cases of ErnA of *Bradyrhizobium* sp. ORS3257 [72], NopD of *Bradyrhizobium* sp. XS1150 and Bel2-5 of *B. elkanii* USDA61 [81,82,83], and even NopL and NopM of *S. fredii* NGR234 [93,97]. Some rhizobial T3Es seem to be phosphorylated by plant kinases in vitro, such as NopP, NopL, and NopM of *S. fredii* NGR234 [92,93,94,95,96,97,100]. More specifically, NopL and NopM are phosphorylated in planta by the mitogen-activated protein kinases (MAPK), salicylic acid-induced protein kinase (SIPK) of *Nicotiana tabacum* (NtSIPK) [93,96,97]. However, only the physical interactions of NopL and NtSIPK within the plant cell nucleus has been described [93]. Some T3E possess specific domains for the modulation of posttranslational modifications, such as ubiquitinylation or SUMOylation, which can influence diverse plant aspects. Those T3E includes NopM of *S. fredii* NGR234, and NopD of *Bradyrhizobium* sp. XS1150 and Bel2-5 of *B. elkanii* USDA61 from the NopD family of T3E, respectively [66,80,83,96,97]. Finally, some T3E possess proteolytic activity, although the nature of this activity seems to be different. In this regard, NopE1 of *B. diazoefficiens* USDA110 and NopT of *S. fredii* NGR234 display autoproteolytic activity, and very interestingly, NopT also cleaves the protein kinase Arabidopsis AvrPphB Susceptible 1 (AtPBS1) and its homologous in soybean, GmPBS1-1, and thus can activate these target proteins [106].

## 4. The Role of the Rhizobial Type 3 Secretion System in Symbiosis

The first evidence of the existence of a rhizobial T3SS was obtained after the confirmation that *S. fredii* USDA257 secreted some proteins to the extracellular milieu upon flavonoids induction. This protein secretion determines the incapacity of this strain to nodulate agronomically improved American soybean varieties [113,114]. Since that time, the role of the T3E in symbiosis has been extensively studied in Meso-, Brady-, and Sinorhizobia by different research groups [8,47,49,64] focusing their research mainly on the symbiotic interaction with soybean, and *Vigna* and *Lotus* species (Table 2). As previously mentioned, in some cases, the recognition of rhizobial T3E by legume plant protein receptors blocks nodulation. This phenotype resembles the gene-for-gene resistance of the phytopathogen–plant relationship, which in the case of rhizobia–legume interaction is translated to the determination of the host specificity [115]. However, the latest findings indicate that other effectors exert exactly the opposite effect: they are essential for nodulation [72,78,82]. The effect of the different T3E on the symbiotic phenotype will vary from beneficial, to neutral, to detrimental and will always depend on the effector studied and the final balance of the different effects of the different effectors on the symbiotic process. In Table 2, a compendium of positive or negative effects of the effectors analyzed to date is shown.

### 4.1. Soybean and Wild Soybeans

*Glycine max* L. Merr. (soybean) cultivation is extended worldwide due to its economical and agronomical importance. In terms of the study of symbiotic signals, soybean is probably the model for the study of T3E function in the rhizobium–legume symbiosis. In fact, all Brady- and Sinorhizobia able to nodulate soybean possess a functional T3SS (but not all rhizobial strains with a T3SS can nodulate soybean). Symbiotic phenotypes vary from nodulation blocking to host-range extension, and even promoting nodulation in the absence of NF (Table 2).

During soybean domestication, many natural phenotypic changes affecting plant development, flowering time, seed size and protein and oil content, among others, have occurred [157]. In this process of domestication, traits controlling the formation of symbiotic root nodules by several host resistance (*R*) genes, referred to as *Rj*/*rj* genes, have been maintained in agronomically improved soybean cultivars. Four Rj genotypes control nodulation in soybean: (i) Rfg1 soybeans restrict nodulation with some *S. fredii* strains such as USDA257, USDA205, and USDA193 [143,158]. The *Rfg1* gene encodes a member of the Toll-interleukin receptor/nucleotide-binding site/leucine-rich repeat class (TIR-NBS-LRR) of plant resistance proteins; (ii) Rj2 soybeans, carrying an allelic variant of *Rfg1*, also restrict nodulation with *B. japonicum* USDA122 [143]; (iii) Rj3 soybeans cannot be nodulated by some *B. elkanii* strains such as USDA33, BLY3-8, or BLY6-1 [159]; and (iv) strains such as *B. japonicum* Is-34 or *B. elkanii* USDA61 cannot nodulate Rj4 soybeans. Intriguingly, the soybean *Rj4* gene codes for a thaumatin-like protein (TLP) that belongs to pathogenesis-related (PR) protein family 5 [160,161,162] and it is not clear yet the mechanisms by which these proteins impede nodulation. In these cases, the recognition of rhizobial T3E by these R proteins would trigger a soybean ETI response to block nodulation [128,163].

The Rj2/Rfg1 protein is the soybean determinant restricting nodulation by some *B. japonicum*, *B. diazoefficiens*, and *S. fredii* strains. Polymorphisms of seven amino acid residues (E452K, I490R, Q731E, E736N, P743S, E756D, and R758S) define three allelic groups of Rj2/Rfg1. Whereas *Rj2*/*rfg1* restricts nodulation with some *B. japonicum* and *B. diazoefficiens* strains, *rj2*/*Rfg1* restricts nodulation with strains of *S. fredii*. On the other hand, *rj2*/*rfg1* allows most *B. japonicum*, *B. diazoefficiens*, and *S. fredii* to form nodules [143,158]. Recent works have associated the incapacity of some rhizobial strains to nodulate certain soybean cultivars with secretion of T3E, mainly with members of the NopP family of effectors. Thus, the incapacity of some *B. japonicum* strains to nodulate Rj2 soybeans is mediated by the rhizobium-specific effector NopP, in which three amino acid residues (R60, R67, and H173) are involved [163]. With respect to the legume partner, it has been reported that only the isolecucine residue (I490) in Rj2 is responsible for the incompatibility [125]. Recently, Rehman et al. [129] found that NopP from *S. fredii* CCBAU25509, but not that from CCBAU45436, is responsible for the incapacity of this strain to nodulate the *rj2*/*Rfg1* allelic group of soybeans. However, there are no clear differences in the sequences of NopP from different *S. fredii* strains able or unable to nodulate these soybeans and no direct interaction between Rfg1 and NopP was found. Very recently, a new soybean *R* gene, *Glycine max* Nodule Number Locus 1 (*GmNNL1*), has been described. GmNNL1 is similar to Rj2 and Rfg1, a TIR-NBS-LRR protein, and interacts with the *B. japonicum* USDA110 NopP to inhibit nodulation during root hair infection [164].

On the other hand, the inability to nodulate Rj4 soybeans is mediated by proteins with a C48 protease domain present in the NopD family of rhizobial effectors [84,127,165]. The catalytic C48-peptidase domain is involved in SUMOylation and de-SUMOylation of host proteins. Small ubiquitin-modifiers (SUMO) are small proteins used by eukaryotic cells to posttranslational modify substrate proteins in a specific manner. These modifications can alter protein stability and activity. While SUMOylation causes the activation/repression of certain transcription factors, de-SUMOylation causes the opposite effect [166]. NopD from *B. japonicum* is delivered to the nucleus of the host cell and can SUMOylate and de-SUMOylate soybean proteins [83]. It is thought that the isopeptidase activity shown by NopD would mimic the activity of a host protease that eliminates SUMO modifications, while the peptidase activity would be necessary to activate SUMO precursors [167].

It is worth mentioning here that there have been described several cases in which rhizobia can effectively nodulate their host plants in the absence of NF, in a process mediated by T3E and carried out through intercellular infection. This is the case for some of the photosynthetic *Bradyrhizobium* strains (those harboring a T3SS) able to nodulate tropical *Aeschynomene* species [23,72] or for a *nodC* mutant of *B. elkanii* USDA61, unable to produce NF, with *G. max* cv. Enrei [133]. The protein responsible for the capacity of *B. elkanii* USDA61 to nodulate soybean in the absence of NF, Bel2-5, possesses the C48 peptidase domain present in the NopD family of effectors. Recent efforts could not determine which *B. elkanii* USDA61 Bel2-5 domains are involved in blocking/inducing nodulation in soybean [81], although the structure and presence of the different NopD domains is similar in other rhizobial species.

Finally, and despite the specific effector involved having not yet been identified, it has been reported that the T3SS of *B. elkanii* BLY3-8 is responsible for the incapacity of this strain to nodulate Rj3 soybeans [131].

*Glycine soja* (Siebold and Zucc.) is the wild ancestor of the domesticated soybean [168]. Studies carried out by Temprano-Vera et al. [144] indicate that inactivation of the T3SS significantly impacts symbiosis with several *G. soja* accessions. Thus, accession CH2 from northern China forms nodules with *S. fredii* NGR234 but not with *S. fredii* HH103. While T3SS mutants of NGR234 lose their capacity to nodulate this accession, HH103 T3SS mutants gain nodulation. Therefore, one or more T3E secreted by NGR234 can promote CH2 nodulation while the combination of Nops secreted by HH103 prevents it. In the case of accession CH3, also from northern China, HH103 T3SS mutants retain nodulation. However, inactivation of the NGR234 T3SS totally abolishes nodulation. The different effect of the T3SS mutations in the symbiosis with the different *G. soja* accessions suggests that these wild soybeans could have different *Rj*/*rj* genotypes or even new *R* genes that could have been transferred to cultivated soybeans during the process of domestication.

### 4.2. Vigna spp.

One of the two most widely consumed types of legumes, together with soybean, belongs to the genus *Vigna*. Among all the species only a very few have been domesticated so far [169,170]. *Vigna unguiculata* L. Walp. (cowpea) is one of the most important grain legume crops in the world with a larger zone of occurrence and cultivation, while *Vigna radiata* L. Wilczek (mung bean) and *Vigna mungo* var. *mungo* L. Hepper (black gram) are consumed and produced especially in Asia [170,171].

Rhizobial T3SS induce positive, neutral, and negative effects in symbiosis with *Vigna* spp. This scenario is well reflected in the *Sinorhizobium*/*Bradyrhizobium*–cowpea symbiotic interactions. However, observations turn more interesting with findings from studies about *Bradyrhizobium* and other *Vigna* spp. symbiotic relationships. In this sense, this plant could be a very good model to study the role of Bradyrhizobial T3E, because their T3SS are responsible for the symbiotic compatibility/incompatibility phenotype in these legumes and because they possess a wide range of T3E.

Regarding the negative impact on symbiosis, rhizobial T3SS can cause a nodulation-blocking phenotype, as occurs in the symbiosis between *B. elkanii* USDA61 and *V. mungo* cv. VM3003 and U-THONG2, *V. aconitifolia* or *V. trinervia* [155]. Impressively, the T3SS renders incompatible the symbiotic interaction between various Bradyrhizobia species with several *V. radiata* cultivars, such as *B. vignae* ORS3257 with CN72, KPS2, SUT1, SUT4, V4758, and V4785; *B. diazoefficiens* USDA110 with KPS2, V4718, and V4785; *Bradyrhizobium* sp. DOA9 with V4718 and V4758; and *B. elkanii* USDA61 with KPS1 [130,135,156]. So far, the genetic basis of how each T3SS is involved in the suppression of nodulation in a specific symbiotic interaction has not been clearly elucidated. Nevertheless, one possibility is that the plant recognizes a specific T3E by a resistance protein inducing a very strong ETI, which finally blocks nodulation. In this regard, it is worth remarking that the T3E InnB from *B. elkanii* USDA61, which causes nodulation restriction in *V. radiata* cv. KPS1, could be recognized specifically by this plant, as well as by *V. aconitifolia*, but not by other *Vigna* spp. or Rj4-soybeans [77,78,130,155]. In line with this background, NopP2 seems to be responsible for the symbiotic incompatibility between *B. vignae* ORS3257 and *V. radiatia* cv. SUT1. This T3E shares 71.9% of identity with NopP from the *B. japonicum* strains USDA122 and USDA110 but does not conserve all amino acids required in USDA122 NopP for Rj2-soybean incompatibility [135]. This finding could be explained by the possibility that *V. radiata* and soybean Rj2 orthologs evolved differently during the rhizobium–plant coevolution process. Thus, plants could monitor the compatibility of their symbionts by the recognition of different specific NopP variations.

On the other hand, some Bradyrhizobial T3SS are also required for a successful nodulation, being involved in host-range extension. This is the case of the symbiotic interactions between *B. diazoefficiens* USDA110 with *V. radiata* cv. CN72, and *B. elkanii* USDA61 with *V. mungo* cv. MASH or IBPGR2775-3 cultivars [155]. Very interestingly, although USDA61 possesses various T3E that are positive for nodulation in these *V. mungo* cultivars, NopL has been found to be a key determinant for the T3SS-triggered nodulation phenotype. Specifically, this T3E is required for the early nodule development in an NF-dependent manner [155]. However, the molecular mechanisms underlying USDA61 NopL mode of action are yet to be elucidated.

### 4.3. Lotus spp.

*Lotus japonicus*, together with *Medicago truncatula*, are the model plants commonly used to study the molecular basis of the symbiotic process due to their small size and the availability of a high variety molecular biology tools. T3E play important and distinct roles in the interaction of *Lotus* spp. with different rhizobial partners. Regarding the natural partner of Lotus, *Mesorhizobium loti*, the T3SS (present in MAFF303077 but not in R7A) may play different roles in symbiosis [85,149]: positive with *L. corniculatus*, *L. filicaulis*, and *L. tenuis* INTA PAMPA; neutral with *L. japonicus* MG-20, and negative with *L. halophilus*, *L. peregrinus*, *L. subbiflorus*, and *L. tenuis* Esmeralda [85,149]. Regarding *S. fredii* strains, NGR234 NopL and NopM have a neutral and a positive role with *L. japonicus* Gifu and MG-20, respectively [92,97], whereas the T3SS have a negative role in the symbiosis of HH103 with *L. burttii*. Interestingly, *S. fredii* HH103 cannot nodulate *L. japonicus* GIFU. However, T3SS mutants gain the capacity to nodulate this legume. When analyzing the effect of the mutation of each T3E secreted by HH103, NopC was revealed to be the one involved in blocking nodulation [148]. With respect to *Bradyrhizobium*, there are several works regarding the role of T3SS in the symbiosis of *Lotus* spp. with *B. elkanii* USDA61 and *Bradyhizobium* sp. SUTN9-2 [89,147]. In general, the T3SS of these *Bradyrhizobium* strains have a negative role in symbiosis with different *Lotus* species. The analysis of the individual effects in symbiosis of the different T3E found NopF of *B. elkanii* USDA61 to be the effector responsible for blocking nodulation with *L. japonicus* Gifu and NopM as the T3E involved in triggering a nodule early-senescence response. Authors suggest that there are at least three checkpoints that control infection and the development of the nodulation process in the different *Lotus* accessions [89].

### 4.4. Aeschynomene spp.

*Aeschynomene* species are tropical plants nodulated by Bradyrhizobial strains. Some interactions were initially considered peculiar cases in Rhizobium–legume symbiosis, since these rhizobia possess a functional T3SS but are incapable of producing NF [23]. Thus, an alternative nodulation process can be developed by an NF-independent pathway in a T3SS-dependent manner. However, growing studies on symbiotic interactions have broken this exception of the rule, due to the occurrence of the symbiotic interaction between *B. elkanii *USDA61 and *G. max* cultivar Enrei, as previously mentioned [133]. In general, the T3SS and its T3E exert neutral or positive effects in the *Bradyrhizobium*-*Aeschynomene* symbioses studied so far (Table 2). Remarkably, *Bradyrhizobium* sp. ORS3257 and *B. elkanii* USDA61 T3SS are strictly required for nodulation in *A. indica* [72,118]. Findings from ORS3257 studies support the notion that T3E play complementary roles in nodulation with *A. indica*. Very interestingly, the T3E ErnA triggers the nodulation process in this legume and, when transferred to *Bradyrhizobium* DOA9, confers on this strain the ability to nodulate this plant and promotes the formation of nodule-like structures on its roots when *ernA* is ectopically expressed [72].

## 5. Conclusions and Perspectives

The findings summarized in this review point to an increasing interest in the identification and determination of the functions of the different rhizobial T3E. Dozens of different legume cultivars have been tested to determine the symbiotic effect of blocking T3SS secretion or the inactivation of single T3E genes. Thus, we have made an exhaustive compilation of all the symbiotic phenotypes published so far and classified them into neutral, positive, or negative effects in symbiosis. In some cases, host-range is extended (positive effects) or nodulation is blocked (negative effect). In this sense, T3E show different facets depending on the host plant and thus resemble Dr. Jekyll and Mr. Hyde in symbiosis.

Due to the important effects on their hosts caused by T3E, efforts are currently made to identify new T3E by proteomics, transcriptomics and bioinformatic analyses. However, functions of the T3E are complex and often simultaneous. Thus, experimental trials focused on a single T3E cannot expose their multiple functions and the interconnections established among them during the symbiotic process. Symbiotic and pathogenic bacteria share several structural and functional similarities in terms of the T3E, even during the bacterium–plant coevolution process, where some T3E are recognized by specific R proteins that block infection/nodulation. The identification and eventual modification of new plant R proteins could be one of the next challenges for extending the use of rhizobia as biofertilizers since their presence in certain species or cultivars permit or abort infection. On the other hand, it is very remarkable that a single mutation in a T3E gene could block/allow nodulation in a host plant, given the complexity of the nodulation process. Thus, it could be considered a highly valuable tool for the selection of broader host-range rhizobial strains used as inoculants [172].

In addition, it is striking that several T3Es are capable of triggering nodulation per se, which provides, together with other findings summarized in this review, new perspectives on tackling plant breeding programs, and even the challenge of extending the nitrogen fixation capacity to other non-legume plants, such as cereals. Up to now, the current knowledge about the effects of T3E on cereals is extremely limited, having only shown that the symbiotic T3SS causes an effect in rice colonization [116,117]. Therefore, more efforts should be made to enrich our knowledge about rhizobial T3SS, and thus creating a future reference for the sustainable agriculture strategies based on the reduction in the use of nitrogen fertilizers.

## Figures and Tables

**Figure 1 ijms-23-11089-f001:**
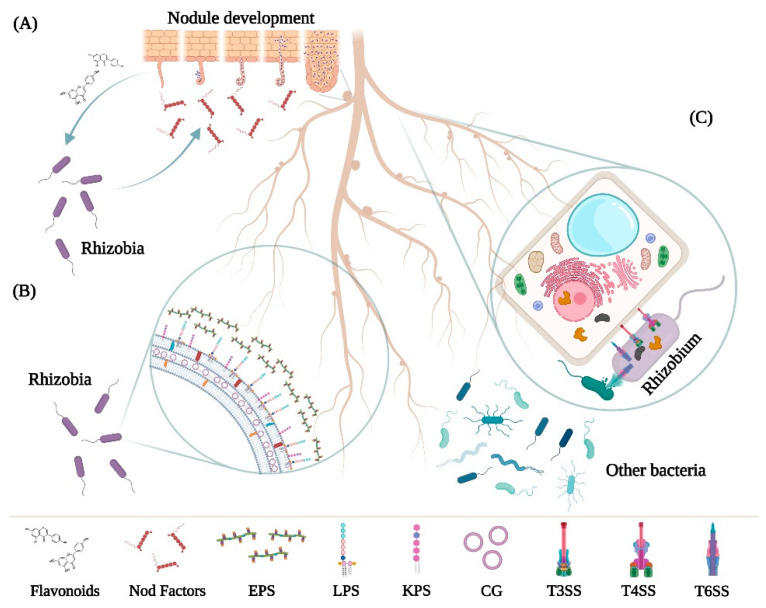
General checkpoints in rhizobia–legumes symbiosis. (**A**) Specificity in nod factor and flavonoid production and detection during rhizobial colonization, infection, and invasion throughout the nodulation process. (**B**) Rhizobial surface polysaccharides [exopolysaccharides (EPS), lipopolysaccharides (LPS), K-antigen polysaccharides (KPS) and cyclin glucans (CG)] play a role either as signal molecules required for the progression of symbiosis and/or as protective agents against plant defense responses. (**C**) Effectors translocated through the rhizobial type III secretion system (T3SS), type IV secretion system (T4SS) or type VI secretion system (T6SS) can modulate plant defense responses to facilitate bacterial infection of the host. The T6SS can also be used as a killing machine devoted to outcompeting other rhizosphere bacteria.

**Table 1 ijms-23-11089-t001:** List of described rhizobial type three secretion system effectors.

T3E	Gene Present in	Domains	Protein Homology	Subcellular Localization	Function/Target	Ref.
ErnA	*Bradyrhizobium* sp. ORS3257	-	Rhizobium-specific	Nucleus (*Bradyrhizobium* sp. ORS3257)	-	[72]
GunA/GunA2	*Sinorhizobium* spp. *Bradyrhizobium* spp. *R. leguminosarum* *M. amorphae*	Glycoside hydrolase family 12 (GH12)	Rhizobium-specific	-	Cellulase (xyloglucan hydrolase) (*S. fredii* HH103 and USDA257 *B. diazoefficiens* USDA110)	[73,74,75,76]
InnB	*Bradyrhizobium* spp. *M. loti* MAFF303099	-	Rhizobium-specific	-	-	[77,78]
NopC ^T^	*Sinorhizobium* spp. *Bradyrhizobium* sp. BRP14	-	Rhizobium-specific	-	-	[79]
**NopD family**						
NopD/Bel2-5/Blr1693/Bll8244/Blr1705	*Sinorhizobium* spp. *Bradyrhizobium* spp.	C48 cysteine protease * /EAR motif/IRS/NLS	XopD (X. *campestris*)/PsvA (*P. syringae*)	Nucleus (*Bradyrhizobium* sp. XS115 and *B. elkanii* USDA61)	SUMOylation/deSUMOylation/AtSUMO1-2, GmSUMO and PvSUMO (*Bradyrhizobium* sp. XS1150)	[66,80,81,82,83]
MA20_12780	*Bradyrhizobium* spp.	C48 cysteine protease	XopD	-	-	[84]
Mlr6316	*Mesorhizobium* spp.	C48 cysteine protease	XopD	-	-	[85]
BRAD325_v2_7792	*Bradyrhizobium* spp.	C48 cysteine protease	XopD	-	-	[70]
NopE1/NopE2	*Bradyrhizobium* spp.	DUF1521	-	-	Reduces phytohormone-mediated ETI-type response (*B. diazoefficiens* USDA110)	[86,87,88]
NopF	*Bradyrhizobium* spp.	Acyl-CoA N-acyltransferase superfamily	HopBG1 (*P. syringae*)	-	-	[89]
NopI	*Sinorhizobium* spp. *Bradyrhizobium* spp. *Microvirga* spp.	-	Rhizobium- specific	-	-	[90]
NopJ ( = Y4lO)	*S. fredii* NGR234 *Bradyrhizobium* spp. *M. amorphae*	C55 cysteine protease Ser/Thr acetyltransferase	YopJ/AvrRxv family	-	-	[66,91]
NopL ^T^	*Sinorhizobium* spp. *Bradyrhizobium* spp.	Ser-pro motifs	Rhizobium-specific	Nucleus (*S. fredii* NGR234)	MAPK substrate (*S*. *fredii* NGR234) SIPK (SA-induced protein kinase) (*S. fredii* NGR234)	[92,93,94,95]
NopM/NopM1 ^T^ /NopM2/NopM3	*Sinorhizobium* spp. *Bradyrhizobium* spp. *Microvirga* spp.	NEL E3-ubiquitin ligase Leucine rich repeat (LRR)	IpaH-like family	Nucleus (*S. fredii* NGR234)	NEL E3 ubiquitin ligase (*S. fredii* NGR234)/ NtSIPK (*S. fredii* NGR234)	[96,97]
NopP ^T^	*Sinorhizobium* spp. *Bradyrhizobium* spp.	-	Rhizobium-specific	Plasma membrane (*M. amorphae* CCNWGS0123)	MAPK substrate (*S*. *fredii* NGR234) TRAPPC13 (*M. amorphae* CCNWGS0123)	[98,99,100,101]
NopT ^T^ /NopT1/NopT2	*Sinorhizobium* spp. *Bradyrhizobium* spp. *Mesorhizobium* spp.	C58 cysteine protease N-term predicted to be myristoylated and palmitoylated in plant cells	AvrPphB (*P. syringae*) YopT (*Yersinia* sp.)	Plasma membrane (*S. fredii* NGR234)	ATP-citrate synthase α chain protein 2/HR-induced response protein from *R. pseudoacacia* (*M. amorphae* CCNWGS0123) Soybean kinase GmPBS1-1 (*S. fredii* NGR234)	[102,103,104,105,106,107,108,109,110]
**Shikimato kinase-like family**						
Mrl6331	*Mesorhizobium* spp. *Bradyrhizobium* spp. *Sinorhizobium* spp.	Shikimato kinase-like	-	-	-	[85,111]
Mlr6361	*Mesorhizobium* spp. *Sinorhizobium* spp.	Shikimato kinase-like		-	-	[111]
Mrl6358	*Mesorhizobium* spp. *Bradyrhizobium* spp. *S. psoraleae*	-	-	-	-	[111]

* Ubiquitin-like protease; (ULP)-like. ^T^ Translocated effector (validated). EAR motif: Ethylene-responsive element-binding factor-associated amphiphilic repression. NLS: Nuclear localization signal. IRS: Internal repeat sequences.

**Table 2 ijms-23-11089-t002:** Symbiotic phenotypes observed due to the presence of a functional T3SS or to secretion of T3SS effectors in different legumes.

Plant			Positive Effect (Including Host-Range Extension)	Neutral Effect	Negative Effect (Including Nodulation-Blocking Phenotype/T3E Involved)	Ref.
** *Legume plants* **						
** *Genera* **	** *species* **	** *cultivar* **				
*Aeschynomene*	*americana*			T3SS (*Bradyrhizobium* sp. SUTN9-2 and DOA9)		[116,117]
	*afraspera*		T3SS (*Bradyrhizobium* ORS285)	T3SS (*Bradyrhizobium* sp. DOA)		[117,118]
	*evenia*			T3SS (*Bradyrhizobium* ORS285)		[118]
	*indica*		**T3SS** and **ErnA** (*B. vignae* ORS3257 and *B. elkanii* USDA61) **NopT** and **NopAB** (Fix*)/NopM1 and NopP1 (*B. vignae* ORS3257)	NopL, NopP2, BRAD3257_v2_7792 (*B. vignae* ORS3257) T3SS (*Bradyrhizobium* ORS285)	NopAO (*B. vignae* ORS3257)	[70,72,118]
	*sensitiva*			T3SS (*Bradyrhizobium* ORS285)		[118]
	*nilotica*			T3SS (*Bradyrhizobium* ORS285)		[118]
	*uniflora*				T3SS (*Bradyrhizobium* ORS285)	[118]
*Amorpha*	*fruticosa*		T3SS (*M. amorphae* CCNWGSO123)			[119]
*Arachis*	*hipogaea*	Thai Nan			T3SS (*Bradyrhizobium* sp. DOA9)	[117]
*Cajanus*	*cajan*		T3SS (*B. diazoefficiens* USDA110)	T3SS (*S. fredii* HH103) GunA and GunA2 (*B. diazoefficiens* USDA110)	T3SS (*S. fredii* USDA191)	[65,73,120]
*Crotalaria*	*juncea*			NopJ, NopL, NopM and NopP (*S. fredii* NGR234) T3SS (*S. fredii* HH103)	**T3SS** (*Bradyrhizobium* sp. DOA9) **T3SS** (Fix*)/NopT (*S. fredii* NGR234)	[65,100,102,105,117]
	*pallida*				NopT (*S. fredii* NGR234)	[106]
*Desmodium*	*tortuosum*			T3SS (*Bradyrhizobium* sp. DOA9)		[117]
*Erythrina*	*variegata*		NopP (*S. fredii* HH103)		**T3SS** (Fix*) (*S. fredii* HH103 and USDA257)	[65,121,122,123]
*Flemingia*	*congesta*		T3SS, NopL, NopP ^†^ and NopX (*S. fredii* NGR234)	NopL and NopP ^†^ (*S. fredii* NGR234)		[65,98,100,124]
*Glycine*	*max*	Akishirome	T3SS (*B. japonicum* USDA122)	MA20_12780 (*B. japonicum* Is-34)		[84,125]
		Aobako (Rj2)			**T3SS** (*B. japonicum* USDA122)	[125]
		Amphor		NopE1 and NopE2 (*B. diazoefficiens* USDA110)		[86]
		Baimaodou	NopD (*S. fredii* HH103)	NopL (*S. fredii* HH103)		[95,126]
		Baipidou		GunA (*S. fredii* HH103)		[76]
		Baoqingheidou		GunA (*S. fredii* HH103)		[76]
		BARC-2 (Rj4)		InnB (*B. elkanii* USDA61)	**T3SS** and **Bel 2-5** (*B. elkanii* USDA61)	[77,81,82,127,128]
		BARC-3 (rj4)	T3SS (*B. elkanii* USDA61)			[128]
		Bayuezha		NopL (*S. fredii* HH103)		[95]
		C08 (Rfg1)			**T3SS** and **NopP** (*S. fredii* CCBAU25509)	[129]
		Charleston	GunA and NopD (*S. fredii* HH103)	NopL (*S. fredii* HH103)		[76,95,126]
		Chidou1	NopD (*S. fredii* HH103)			[126]
		Chizuka Ibaraki 1 (Rj2)			**T3SS** (*B. japonicum* USDA122)	[125]
		Clark (rj1)	**T3SS** (*B. elkanii* USDA61)			[130]
		COL/Ehime/1983/ Utsunomiya 37		T3SS (*B. japonicum* USDA122)		[125]
		Danzhidou	GunA (*S. fredii* HH103)			[76]
		Date Cha Mame (Rj2)			**T3SS** (*B. japonicum* USDA122)	[125]
		D51 (Rj3)			**T3SS** (Fix*) (*B. elkanii* BLY3-8)	[131]
		Dongnong594	NopD (*S. fredii* HH103)	NopL and GunA (*S. fredii* HH103)		[76,95,126]
		EMBRAPA-48	T3SS (*B. elkanii* SEMIA587)			[132]
		En1282 (Nfr1)	**T3SS** and **Bel 2-5** (*B. elkanii* USDA61)			[82]
		Enrei	T3SS (*B. elkanii* USDA61)			[133]
		Fengdihuang	GunA (*S. fredii* HH103)			[76]
		Hill (Rj4)			**T3SS** (*B. elkanii* USDA61)	[130]
		Fukuyutaka (Rj4)			**MA20_12780** (*B. japonicum* Is-34)	[84]
		Hardee (*Rj2*)	T3SS (*B. vignae* ORS3257)	NopP2 (*B. vignae* ORS3257)	**T3SS** (*B. japonicum* USDA122)	[134,135]
		Heidou		NopL (*S. fredii* HH103)		[95]
		Heihe 13	GunA (*S. fredii* HH103)			[76]
		Heinong 33	T3SS (*S. fredii* HH103)			[122]
		Heinong 35	NopD (*S. fredii* HH103)			[126]
		Himeshirazu		T3SS (*B. japonicum* USDA122)		[125]
		Huangpishanzibai		NopL (*S. fredii* HH103)		[95]
		JD17			**T3SS** and **NopP** (*S. fredii* CCBAU25509 and CCBAU83666)	[136]
		Jihei 4		GunA (*S. fredii* HH103)		[76]
		Kenjian28	NopD (*S. fredii* HH103)			[126]
		Kochi	T3SS (*S. fredii* HH103)			[122]
		Kumaji 1 (Rj2)			**T3SS** (*B. japonicum* USDA122)	[125]
		Kurakake 1 (Rj2)			**T3SS** (*B. japonicum* USDA122)	[125]
		Lee (rj2)		T3SS and NopP2 (*B. vignae* ORS3257)		[135]
		Maetsue Zairai 90B (Rj2)			**T3SS** (*B. japonicum* USDA122)	[125]
		Mancangjin		GunA (*S. fredii* HH103)		[76]
		Peking	T3SS (*S. fredii* HH103) NopA (*S. fredii* USDA257) T3SS (*B. elkanii* SEMIA587)	T3SS (*S. fredii* NGR234 and USDA257)	NopB (*S. fredii* USDA257)	[65,121,122,132,137,138]
		Qingdou	NopD (*S. fredii* HH103)		NopL (*S. fredii* HH103)	[95,126]
		Qingpi		GunA (*S. fredii* HH103)		[76]
		Shakkin Nashi		T3SS (*B. japonicum* USDA122)		[125]
		SN14	T3SS and GunA (*S. fredii* HH103)			[139]
		Suinong14	NopD (*S. fredii* HH103)	NopL (*S. fredii* HH103)		[95,126]
		Suinong 15	GunA (*S. fredii* HH103)			[76]
		Tokachi Nagaha		T3SS (*B. japonicum* USDA122)		[125]
		Tribune		T3SS (*S. fredii* HH103)		[122]
		Wanhuangdadou		NopL (*S. fredii* HH103)		[95]
		Williams 82 and McCall (Rfg1/rj2)	T3SS, GunA, NopC and NopI (*S. fredii* HH103)	T3SS (*B. elkanii* SEMIA587)	**T3SS**, **NopA** and **NopB** (*S. fredii* USDA257) NopL and NopP (*S. fredii* HH103)	[60,65,79,121,122,123,132,137,138,140,141,142,143]
		Zheng9525	NopD (*S. fredii* HH103)			[126]
		ZYD00006	NopD and NopL (*S. fredii* HH103)			[95,126]
	*soja*	*Rj2* (JP90448, JP9052, JP231394, JP231659)			**T3SS** (*B. japonicum* USDA122)	[125]
		*rj2* (JP110740, JP231372, JP231659)		T3SS (*Bradyrhizobium* sp. DOA9)		[125]
		*rj2* (JP233152)	T3SS (*B. japonicum* USDA122)			[117]
		CH2	**T3SS** (*S. fredii* NGR234)		**T3SS** (*S. fredii* HH103)	[144]
		CH3	**T3SS** (*S. fredii* NGR234) T3SS (*S. fredii* HH103)			[144]
		CH4	T3SS (*S. fredii* HH103 and NGR234)			[144]
*Glycyrrhiza*	*uralensis*		T3SS (*S. fredii* HH103)			[122]
*Indigofera*	*tintorea*			T3SS (*Bradyrhizobium* sp. DOA9)		[117]
*Lablab*	*purpureus*		**T3SS**, NopM, NopP and **NopX** (*S. fredii* NGR234)	NopL and NopT (*S. fredii* NGR234)	NopJ (*S. fredii* NGR234)	[96,105,124]
*Leucaena*	*leucocephala*			T3SS, NopA and NopL (*S. fredii* NGR234) Mlr6361 (*M. loti* MAFF303099)	Atypical T3SS (*C. taiwanensis* LMG19424) **T3SS** and **Mlr6316** (*M. loti* MAFF303099)	[52,65,111,145,146]
*Lotus*	*burttii*			T3SS, GunA, NopC, NopD, NopI, NopL, NopM, NopP and NopT (*S. fredii* HH103)	**T3SS** and **NopM**/**NopF** (Fix*) (*B. elkanii* USDA61) T3SS (*Bradyrhizobium* sp. SUTN9-2)	[89,147,148]
	*corniculatus*	*frondosus*	T3SS (*M. loti* MAFF303099)	Mlr6316, Mlr6331, Mlr6358 and Mlr6361 (*M. loti* MAFF303099)		[111,149]
	*filicaulis*		T3SS (*M. loti* MAFF303099)			[149]
	*halophilpus*			Mlr6316, Mlr6331 and Mlr6358 (*M. loti* MAFF303099)	T3SS and Mlr6361 (*M. loti* MAFF303099)	[111,149]
	*japonicus*	Gifu		NopM (*B. elkanii* USDA61) NopD, NopI, NopM and NopT (*S. fredii* HH103) NopL (*S. fredii* NGR234)	**T3SS** (Fix*) (*B. elkanii* USDA61 and 14k062) **NopF** (Fix*) (*B. elkanii* USDA61) T3SS (*Bradyrhizobium* sp. SUTN9-2) **T3SS** and **NopC**/**GunA**, **NopL** and **NopP** (Fix*) (*S. fredii* HH103)	[89,92,147,148]
		MG-20		T3SS (*M. loti* MAFF303099)	**T3SS** and **NopM** (Fix*) (*B. elkanii* USDA61)	[85,89,97]
		Miyakojima	NopF (*B. elkanii* USDA61)		**T3SS** (*Bradyrhizobium* sp. SUTN9-2) T3SS and NopM (*B. elkanii* USDA61)	[147]
	*peregrinus*	carmeli			T3SS (*M. loti* MAFF303099)	[149]
	*subbiflorus*				T3SS (*M. loti* MAFF303099)	[149]
	*tenuis*	INTA Pampa	T3SS (*M. loti* MAFF303099)			[85]
		Esmeralda		Mrl6316 (*M. loti* MAFF303099)	**T3SS** (*M. loti* MAFF303099)	[85]
*Macroptiluim*	*artropurpureum*		T3SS (*B. elkanii* USDA61)	NopE1 and NopE2 (*B. diazoefficiens* USDA110) T3SS (*B. elkanii* SEMIA587)	T3SS (*Bradyrhizobium* sp. SUTN9-2 and DOA9) NopB (*S. fredii* USDA257)	[86,116,117,130,132,138]
*Mimosa*	*pudica*			Atypical T3SS (*C. taiwanensis* LMG19424)		[146]
*Pachyrhizus*	*tuberosus*		**NopP** ^†^ and NopX (*S. fredii* NGR234)	NopJ, NopL, NopP ^†^ and NopT ^†^ (*S. fredii* NGR234)	**T3SS** (Fix*), NopA, NopB, NopM and NopT ^†^ (*S. fredii* NGR234)	[98,100,105,124,142,145,150,151,152]
*Phaseolus*	*vulgaris*		NopT (*S. fredii* NGR234)	T3SS (*S. fredii* NGR234)	NopL (*S. fredii* NGR234)	[93,102,153]
*Robinia*	*hispida*		T3SS (*M. amorphae* CCNWGSO123)	NopP (*M. amorphae* CCNWGSO123)		[101,119,154]
	*pseudoacacia*		T3SS (*M. amorphae* CCNWGSO123)			[119]
*Sophora*	*japonica*			T3SS (*M. amorphae* CCNWGSO123)		[119]
	*xanthantha*			T3SS (*M. amorphae* CCNWGSO123)		[119]
*Stylosantes*	*hamata*		**T3SS** (*Bradyrhizobium* sp. DOA9)			[117]
*Tephrosia*	*vogelii*		T3SS, NopA, NopB, NopP, NopT and NopX (*S. fredii* NGR234)	NopL, NopM and NopJ (*S. fredii* NGR234)	NopD (*Bradyrhizobium* sp. XS1150)	[65,83,100,102,105,124,145,150,151,152]
*Vigna*	*aconitifolia*				**T3SS** and **InnB** (*B. elkanii* USDA61)	[155]
	*angularis*		T3SS (*B. elkanii* USDA61)		InnB (*B. elkanii* USDA61)	[155]
	*mungo*	cv. PI173934	T3SS, InnB, NopL and NopP2 (*B. elkanii* USDA61)	Bel2-5 and NopP1 (*B. elkanii* USDA61)		[78,155]
		MASH	**T3SS** and **NopL**/Bel2-5 and NopP2 (*B. elkanii* USDA61)	InnB and NopP1 (*B. elkanii* USDA61)		[155]
		IBPGR2775-3	**T3SS** and **NopL**/Bel2-5 and NopP2 (*B. elkanii* USDA61)	NopP1 (*B. elkanii* USDA61)	InnB (*B. elkanii* USDA61)	[155]
		MAFF2002M3	T3SS and InnB (*B. elkanii* USDA61)			[155]
		OSUM745	T3SS (*B. elkanii* USDA61)	InnB (*B. elkanii* USDA61)		[155]
		VM3003		InnB (*B. elkanii* USDA61)	**T3SS** (*B. elkanii* USDA61)	[155]
		U-THONG2		InnB (*B. elkanii* USDA61)	**T3SS** (*B. elkanii* USDA61)	[155]
		CQ5785			T3SS and InnB (*B. elkanii* USDA61)	[155]
		-	T3SS and NopP2 (*B. vignae* ORS3257)	Brad7238, ErnA, NopBW and NopL (*B. vignae* ORS3257)		[135]
	*trinervia*			InnB (*B. elkanii* USDA61)	**T3SS** (*B. elkanii* USDA61)	[155]
	*radiata*	CN36		T3SS *(B. elkanii* USDA61)		[130]
		CN72	T3SS (*Bradyrhizobium* sp. SUTN9-2) **T3SS** (*B. diazoefficiens* USDA110)		**T3SS** (*B. vignae* ORS3257) T3SS (*Bradyrhizobium* sp. DOA9)	[156]
		KPS1			**T3SS** and **InnB** (*B. elkanii* USDA61)	[77,78,130,155]
		KPS2	NopE1 and NopE2 (*B. diazoefficiens* USDA110)		**T3SS** (*B. vignae* ORS3257 and *B. diazoefficiens* USDA110) T3SS (*Bradyrhizobium* sp. SUTN9-2 and DOA9)	[86,156]
		SUT1		Brad7238, ErnA, NopBW, NopL and NopP1 (*B. vignae* ORS3257)	**T3SS** and **NopP2** (*B. vignae* ORS3257)	[135]
		SUT4		T3SS (*B. diazoefficiens* USDA110)	**T3SS** (*B. vignae* ORS3257) T3SS (*Bradyrhizobium* sp. SUTN9-2 and *Bradyrhizobium* sp. DOA9)	[116,117,156]
		V4718		T3SS (*Bradyrhizobium* sp. SUTN9-2)	**T3SS** (*B. vignae* ORS3257 and DOA9 and *B. diazoefficiens* USDA110)	[156]
		V4758		T3SS (*Bradyrhizobium* sp. SUTN9-2 and DOA9 and *B. diazoefficiens* USDA110)	**T3SS** (*B. vignae* ORS3257)	[156]
		V4785		T3SS (*Bradyrhizobium* sp. SUTN9-2)	**T3SS** (*B. vignae* ORS3257 and DOA9 and *B. diazoefficiens* USDA110)	[156]
		-		GunA and GunA2 (*B. diazoefficiens* USDA110)		[73]
	*unguiculata*		T3SS and NopAB (*B. elkanii* USDA61) T3SS and NopT (*B. vignae* ORS3257) T3SS, NopC and NopI (*S. fredii* HH103)	Brad7238, Brad7707, ErnA, NopAO, NopBW, NopL and NopP2 (*B. vignae* ORS3257) T3SS, NopA, NopB and NopP (*S. fredii* NGR234) NopB (*S. fredii* USDA257)	InnB (*B. elkanii* USDA61) T3SS (*B. elkanii* SEMIA587) GunA, NopA, NopL (*S. fredii* HH103)	[65,98,132,135,137,138,145,151,155]
** *Non legume plants* **						
*Oryza*	*sativa* L. ssp. *indica*	Phatum Thani 1	T3SS (*Bradyrhizobium* sp. SUTN9-2)-colonization	T3SS (*Bradyrhizobium* sp. DOA9) T3SS (*Bradyrhizobium* sp. SUTN9-2)-growth promotion		[116,117]
	*sativa* L. ssp. *japonica*	Nipponbare		T3SS (*Bradyrhizobium* sp. SUTN9-2)-growth promotion	T3SS (*Bradyrhizobium* sp. SUTN9-2)-colonization	[116]

In bold: Nodulation gaining or nodulation-blocking phenotypes. Fix*: Mutants induce uninfected nodules. ^†^: Contradictory results.

## Data Availability

Not applicable.

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
