# Peer review of "The Rhizobial Type 3 Secretion System: The Dr. Jekyll and Mr. Hyde in the Rhizobium–Legume Symbiosis"

_ijms, 2022, doi:10.3390/ijms231911089_

Round 1

Reviewer 1 Report

The manuscript titled “The rhizobial type 3 secretion system: the Dr. Jekyll and Mr. Hyde in symbiosis” is devoted to reviewing the role of T3SS as molecular basis of a symbiotic association between rhizobia and legumes. This process is regulated with several checkpoints to allow the progression of the symbiosis or its restriction. One of the symbiotic signals for nodulation are the proteins, called effectors, secreted through the type 3 secretion system wide-spread through pathogenic proteobacteria. This secretion system is not present in every rhizobia but its role seems to be essential for several studied symbiotic associations. In this review, the Authors compiled the currently available information about the effect on plant symbiotic phenotypes of different rhizobial effectors. These phenotypes are diverse and highlight the importance of T3SS secretion system in certain rhizobium-legume symbiotic associations.

Despite the high importance of this review and good presentation and discussion of the analyzed literature, the manuscript can be improved in  some parts.

1)    Three parts of the manuscript: part 2) "Rhizobial Nod factors are key signal molecules in most rhizobia-legume symbiotic interactions"; part 3) "Surface polysaccharides also play important roles in rhizobia-legume symbiotic interactions"; and part 4) "Some rhizobia can deliver effector proteins into their hosts through bacterial secretion system" describe in brief some general information about biochemical and molecular mechanisms of rhizobia specificity to host plants, but these parts are not important for the main subject of this review to be separated, and can be shortened and merged with the Part 1 " Introduction".

2)    The manuscript is referring to other reviews 8 times as (reviewed by [**]), and in total 26 reviews are included in the References list (of 172 in total). Would it be better to cite the original research papers instead of sending reader to other reviews? Fragments like this “While in the case of bacterial pathogens the role of these protein secretion systems is generally clear, for beneficial bacteria the function of such kind of systems remains in some cases elusive,  from the point of view of a plant that, in some cases, can even block the invasion of a  bacterial partner that lastly will improve its host fitness (reviewed by [10,11,51,57])” looks confusing. Was this statement repeated in all 4 publications? Is it worth to be repeated again? 

3)    In the part 7) “Type 3 secretion system effectors”, Table 1 (List of described rhizobial type three secretion system effectors)  describs several T3SS effectors in different symbiotic bacteria, but, there is no analysis made for specific bacterial repertoires and core sets of T3SS effectors that is  ususal for analysis of T3SS effector role in plant pathogenic bacteria. Comparison between T3SS effectors in plant pathogens and symbiotic bacteria would be useful for this discussion.

4)    Part 9. "Conclusions and perspectives" unfortunately look too short and too general.  The main suggestion that “The identification of new R proteins could be  one of the next challenges for the future“ (and their sequences) is not enough significant to be called “perspectives”.

Author Response

We would like to thank the comments from reviewers 1 and 2. We believe that their suggestions will clarify and improve the content of the review. We have tried to solve all the concerns indicated by reviewers and therefore we have included some modifications in the manuscript. These modifications are indicated below.

Reviewer 1

The manuscript titled “The rhizobial type 3 secretion system: the Dr. Jekyll and Mr. Hyde in symbiosis” is devoted to reviewing the role of T3SS as molecular basis of a symbiotic association between rhizobia and legumes. This process is regulated with several checkpoints to allow the progression of the symbiosis or its restriction. One of the symbiotic signals for nodulation are the proteins, called effectors, secreted through the type 3 secretion system wide-spread through pathogenic proteobacteria. This secretion system is not present in every rhizobia but its role seems to be essential for several studied symbiotic associations. In this review, the Authors compiled the currently available information about the effect on plant symbiotic phenotypes of different rhizobial effectors. These phenotypes are diverse and highlight the importance of T3SS secretion system in certain rhizobium-legume symbiotic associations.

Despite the high importance of this review and good presentation and discussion of the analyzed literature, the manuscript can be improved in some parts.

1) Three parts of the manuscript: part 2) "Rhizobial Nod factors are key signal molecules in most rhizobia-legume symbiotic interactions"; part 3) "Surface polysaccharides also play important roles in rhizobia-legume symbiotic interactions"; and part 4) "Some rhizobia can deliver effector proteins into their hosts through bacterial secretion system" describe in brief some general information about biochemical and molecular mechanisms of rhizobia specificity to host plants, but these parts are not important for the main subject of this review to be separated, and can be shortened and merged with the Part 1 " Introduction".

Thank you very much for your comment. Following your suggestion and that of reviewer 2, we have rearranged the first parts of the manuscript. Thus, the first five parts have been merged into one (“Introduction”).

2) The manuscript is referring to other reviews 8 times as (reviewed by [**]), and in total 26 reviews are included in the References list (of 172 in total). Would it be better to cite the original research papers instead of sending reader to other reviews? Fragments like this “While in the case of bacterial pathogens the role of these protein secretion systems is generally clear, for beneficial bacteria the function of such kind of systems remains in some cases elusive, from the point of view of a plant that, in some cases, can even block the invasion of a bacterial partner that lastly will improve its host fitness (reviewed by [10,11,51,57])” looks confusing. Was this statement repeated in all 4 publications? Is it worth to be repeated again?

Thank you very much for your comment. We have tried to use reviews mainly for the description of general processes in symbiosis (which would require the citation of a huge number of original articles) and avoid as much as possible in the description of the T3E.

Concerning the fragment of text mentioned by the reviewer, we agree that it is confusing. Therefore, we have deleted most of it. In any case, the statement “for beneficial bacteria, the function of such kind of systems remains in some cases elusive, from the point of view of a plant that, in some cases, can even block the invasion of a bacterial partner that lastly will improve its host fitness” is fully justified in the description of the symbiotic effect of all the rhizobial T3E mentioned in Table 2.

3) In the part 7) “Type 3 secretion system effectors”, Table 1 (List of described rhizobial type three secretion system effectors) describes several T3SS effectors in different symbiotic bacteria, but there is no analysis made for specific bacterial repertoires and core sets of T3SS effectors that is ususal for analysis of T3SS effector role in plant pathogenic bacteria. Comparison between T3SS effectors in plant pathogens and symbiotic bacteria would be useful for this discussion.

We agree with reviewer 1 in that in this review there is not an analysis of the different rhizobial T3E repertoire and core sets of effectors. This analysis was exhaustively and remarkably made by Tampakaki et al. in 2014. Several recent reviews about the rhizobial T3SS have been somehow focused on the biochemical properties of the T3E and have complemented this original analysis (Teulet et al. 2022). However, we decided to write something different to add novelty to the review. In this sense, we found that there was a general statement that symbiotic phenotypes associated to the rhizobial T3SS could be neutral, positive, or negative but there was not a compilation of the recent results published in an enormous number of legume species and cultivars. In addition, we also found that research groups that have in the last years incorporated to the study of the rhizobial T3SS do not include “old” results in their discussions. There was a problem of space to include both analyses and hence we decided to focus on the symbiotic effects. We believe that this review can be a useful guide to help research groups select the legume cultivar that could better fit their symbiotic and molecular analyses.

4) Part 9. "Conclusions and perspectives" unfortunately look too short and too general.  The main suggestion that “The identification of new R proteins could be one of the next challenges for the future (and their sequences)” is not enough significant to be called “perspectives”.

We agree with reviewer 1, and also with reviewer 2, in that this section could be clearly improved. Therefore, the section “Conclusions and perspectives” has been completely rewritten.

Reviewer 2 Report

The topic of the review is relevant and is of scientific interest for study of symbiotic relationships between rhizobia and bean plants. The authors processed a large layer scientific papers (|72 publications). The authors made a very interesting overview of the secretory systems, which will be useful both to specialists in this field and to those researchers of symbiosis who are little aware of the importance of these systems in establishing of symbiotic association rhizobia with legumes. It seems to me that this review can be published without major changes. May be the review needs to be slightly modified in construction (but I do not insist) and corrected in some details :

1. The title of the review needs explanations in the introduction and conclusion;

2. In the summary, put more emphasis on the description of secretory systems and their role in symbiosis;

3. The purpose of this review is not stated in the introduction;

4. Paragraphs 2 and 3 can be combined, shortened and even incorporated in the introduction;

5. Paragraphs 4 and 5 can also be combined. Then the emphasis on the T3SS system will be more obvious (paragraphs 6-8);

6. The conclusion does not reflect the relevance and novelty of this review, compared to current literature;

Some remarks:

343....    paragraphs iii and iv are inconsistent with paragraphs i and ii (line                   338) where it was said about genes "Four R genes have been                           traditionally considered in soybean:..."

334...378    please check the designations of genes and proteins

372...378    a brief explanation of the term "SUMOylation" needed 

Author Response

We would like to thank the comments from reviewers 1 and 2. We believe that their suggestions will clarify and improve the content of the review. We have tried to solve all the concerns indicated by reviewers and therefore we have included some modifications in the manuscript. These modifications are indicated below.

Reviewer 2

The topic of the review is relevant and is of scientific interest for study of symbiotic relationships between rhizobia and bean plants. The authors processed a large layer scientific paper (172 publications). The authors made a very interesting overview of the secretory systems, which will be useful both to specialists in this field and to those researchers of symbiosis who are little aware of the importance of these systems in establishing of symbiotic association rhizobia with legumes. It seems to me that this review can be published without major changes. May be the review needs to be slightly modified in construction (but I do not insist) and corrected in some details:

  1. The title of the review needs explanations in the introduction and conclusion.

The title has been changed and he have included the explanation to the title in the abstract and in “conclusions and perspectives” sections.

  1. In the summary, put more emphasis on the description of secretory systems and their role in symbiosis.

The abstract has been completely rewritten.

  1. The purpose of this review is not stated in the introduction.

Thank you for the comment. We have introduced the following sentence in the last paragraph before Figure 1: As we will briefly describe below, different rhizobial molecular signals participate in symbiosis: Nod factors, surface polysaccharides and effector proteins delivered by specialized secretion systems (Figure 1). However, this review focuses on the effector proteins delivered by rhizobial type 3 secretion systems (T3E) and provide a description of the different T3E proteins described so far as well as exhaustive compilation of their role (positive, neutral, or negative) in symbiosis with specific counterparts.  

  1. Paragraphs 2 and 3 can be combined, shortened, and even incorporated in the introduction.

Thank you for your comment. Following your suggestion and that of reviewer 2, we have rearranged the first parts of the manuscript. Thus, the first five parts have been merged into one (Introduction).

  1. Paragraphs 4 and 5 can also be combined. Then the emphasis on the T3SS system will be more obvious (paragraphs 6-8).

Thank you for your comment. Following your suggestion and that of reviewer 2, we have rearranged the first parts of the manuscript. Thus, the first five parts have been merged into one (“Introduction”).

  1. The conclusion does not reflect the relevance and novelty of this review, compared to current literature.

We agree with reviewer 2, and also with reviewer 1, in that this section could be clearly improved. Therefore, the section “Conclusions and perspectives” has been completely rewritten.

Some remarks:

343.... paragraphs iii and iv are inconsistent with paragraphs i and ii (line                   338) where it was said about genes "Four R genes have been                           traditionally considered in soybean:…"

The text has been changed to clarify soybean genotypes.

334...378    please check the designations of genes and proteins

The text in section 4.1 has been checked for protein and gene names designations

372...378    a brief explanation of the term "SUMOylation" needed

The explanation has been included as indicated.

Round 2

Reviewer 1 Report

The manuscript titled "The rhizobial type 3 secretion system: the Dr. Jekyll and Mr. Hyde in the rhizobium-legume symbiosis" reviews recent achievements in the study of rhizobium-legume symbiosis from the point of type 3 secretion system involvement in interaction prosess. The manuscript has been improved considerably and can be published in its present form.